# Comparison of LncRNA Expression Profiles during Myogenic Differentiation and Adipogenic Transdifferentiation of Myoblasts

**DOI:** 10.3390/ijms20153725

**Published:** 2019-07-30

**Authors:** Renli Qi, Xiaoyu Qiu, Yong Zhang, Jing Wang, Qi Wang, Min Wu, Jinxiu Huang, Feiyun Yang

**Affiliations:** 1Animal Nutrition Institute, Chongqing Academy of Animal Science, Rongchang, Chongqing 40240, China; 2Key Laboratory of Pig Industry Sciences, Ministry of Agriculture, Rongchang, Chongqing 402460, China; 3School of Life Science and Engineering, Southwest University of Science and Technology, Mianyang 621010, China

**Keywords:** transdifferentiation, adipogenesis, lncRNA, myoblasts, RNA sequencing, C2C12 cells

## Abstract

Myoblasts could transdifferentiate into adipocytes or adipocyte-like cells, which have the capability of producing and storing intracellular lipids. Long-chain non-coding RNAs (lncRNAs) have many important physiological functions in eukaryotes, which include regulating gene expression, chromosome silencing, and nuclear transport. However, changes in the expression of lncRNAs in muscle cells during adipogenic transdifferentiation have not been investigated to date. Here, C2C12 myoblasts were seeded and then induced to undergo myogenic and adipogenic transdifferentiation. The expression profiles of lncRNAs in various differentiated cells were analyzed and then compared by digital gene expression (DGE) RNA sequencing. A total of 114 core lncRNAs from 836 differentially expressed lncRNAs in adipogenic cells were identified. Further investigation by in silico analysis revealed that the target genes of core lncRNAs significantly enriched various signaling pathways that were related to glucose and lipid metabolism and muscle growth. The *lncRNA-GM43652* gene was a potential regulator of adipogenesis in muscle cells. It showed the highest levels of expression in adipogenic cells, and the knocking down *lncRNA-GM43652* negatively influenced lipid deposition in transdifferentiated myoblasts. This study has identified the novel candidate regulators that may be assessed in future molecular studies on adipogenic conversion of muscle cells.

## 1. Introduction

Transdifferentiation is the transformation of one cell type to another. Myoblasts have the potential to transdifferentiate into adipocytes or adipocyte-like cells under specific induction conditions (i.e., drug stimulation, cytokine treatment) [1,2,3,4,5,6]. Skeletal muscle satellite cells in humans and animals retain broad differentiation capacity, including the generation of adipocytes in an adipogenic medium in a process that the insulin sensitizer—rosiglitazone—enhances [7]. Myogenic C2C12 cells also can be converted to adipocytes by overexpression of PPARγ and C/EBPα, which are two critical mediators for cellular adipogenesis [1,8]. A few skeletal muscle cells could be replaced by fat cells during regeneration in humans [9]. During skin wound healing in mice, the regeneration of hair follicles is accompanied by the transformation of numerous myofibers (myofibroblasts) into white adipocytes [10]. The BMP signaling pathway is activated, which in turn drives the transformation of “myofibroblast-adipocytes” [10].

Adipogenic transdifferentiated muscle cells lose their myogenic ability, but they gain the adipogenic ability to produce and store lipids within cells. The conversion usually occurs in muscle progenitor or satellite cells [3]. The multinucleated myocytes (myotubes) at terminal differentiation do not have the ability to transdifferentiate. The shift between muscle and adipose tissues results in changes in organ function and metabolism. However, to date, the molecular mechanism underlying the conversion of muscle-adipose is not fully understood.

In recent years, non-coding RNAs (ncRNAs) have become a research topic of interest in the life sciences. Various ncRNAs (e.g., miRNAs, ceRNAs, and lncRNAs) are essential for gene transcription, cleavage, translation, modification, and expression, and therefore exhibit a wide range of regulatory roles in different life events [11,12,13]. Some coding genes and non-coding genes (particularly miRNAs) are involved in the control of adipogenic transdifferentiation of myocytes [14,15,16]. However, most of the ncRNAs in humans and animals are long-chain ncRNAs (lncRNAs), with lengths of more than 200 nt. LncRNAs commonly have complex molecular structures and diverse functions, such as the control of cell fate decision, cycle, proliferation, differentiation, and apoptosis [17,18,19]. Several lncRNAs have been known for their important regulatory roles in the formation and physiological function of muscle and fat tissues, such as *LncMyoD*, *Linc-YY1*, *Blnc1*, and *Lnc-ORA* [20,21,22,23,24]. For instance, the expression of the lncRNA *slincRAD* is unregulated during the early differentiation stages of 3T3-L1 adipocytes. The abolishment of the interaction between slincRAD and the DNMT1 gene will result in defective epigenetic regulation, which in turn compromises adipogenesis [25]. However, the expression and function of lncRNAs in the adipogenic transdifferentiation of muscle cells have not been investigated to date.

In this study, we aimed to identify the lncRNAs that are involved in the adipogenic transformation of muscle cells to reveal the molecular mechanism of transdifferentiation. C2C12 is a murine myoblasts cell line that has been used as a classic cell model for the study of muscle growth, development, and function [26]. Here, C2C12 cells were induced to undergo myogenic differentiation and adipogenic transdifferentiation. Subsequently, the expression profiles of lncRNAs in the various differentiated cells were assessed by RNA-seq-based digital gene expression (DGE-Seq) analysis [27,28]. A total of 114 significant differentially expressed (DE) lncRNAs (RPKM > 100, fold-change > 2, and *P*-value < 0.001) between the myogenic cells and the adipogenic cells were obtained, and their target mRNAs were predicted for functional analysis. Finally, a novel lncRNA gene, *lncRNA-Gm43652*, which exhibited the highest expression in adipogenic differentiated cells, was determined to promote adipogenesis in muscle cells. These results provide useful information that may be utilized in the elucidation of the molecular mechanism of adipogenic transdifferentiation of muscle cells.

## 2. Results

### 2.1. The Adipogenic Transdifferentiation of C2C12 Myoblasts

Figure 1A shows that the cells are spirally arranged and they gradually fuse into multinuclear cells with the normal myogenic induction, and then into formatted myotubes after six days of induction. The adipogenic transdifferentiated cells gradually became round via adipogenic induction, with numerous small lipid droplets inside the cells. qRT-PCR detected the key regulatory factors that are related to myogenesis or adipogenesis in different cells (Figure 1B,C). This observation is in line with the fact that *FAS* and *PPARγ*, which are two key adipogenesis markers, exhibit upregulated mRNA expression levels in adipogenic differentiated cells (ADCs) as compared to undifferentiated cells (UDCs) during cellular adipogenesis. The myogenesis regulators, *Myf5* and *myogenin*, showed stable high expression in the myogenic differentiated cells (MDCs). However, the number and size of lipid droplets in the adipogenic myoblasts were relatively fewer and smaller than those of mature adipocytes at the same differentiation induction time (Appendix A). The brown adipocyte-specific genes, *UCP1* and *PRDM16*, both showed higher expression levels in the ADCs when compared to the 3t3-L1 adipocytes (Appendix A). This suggests that the adipogenic cells that had transformed from myoblasts possess the brown adipocyte signatures.

### 2.2. DGE-RNA Sequencing of lncRNAs

High-throughput DGE-RNA sequencing was used to assess the expression of lncRNAs and protein-coding RNAs in UDCs, ADCs, and MDCs to reveal the differential expression of lncRNAs among various cells. Total RNA was extracted from the cells for preparation of RNA libraries. Three RNA sequencing libraries were sequenced on the Illumina^®^ HiSeq 2500 platform. Cufflinks and Scripture assembled a total of 48,615 known protein-coding gene transcripts (for 17,568 mRNAs) and 8594 known lncRNA transcripts (for 6363 lncRNAs) by comparing the clean sequencing data with the current mouse genome.

Figure 2A shows the holistic expression levels of mRNAs and lncRNAs in different cells. The average expression level of the lncRNAs was slightly higher in the differentiated cells relative to the undifferentiated cells. Additionally, no clear difference in the average expression level between lncRNAs and mRNAs was observed. Transcript types and the chromosomal distribution of lncRNAs were determined (Figure 2B,C). The comparison of transcript length and exon number of the two transcripts are shown in Figure 2D,E. Figure 2F shows that most of the lncRNAs contained a relatively shorter ORF when compared with that of the mRNAs.

Low-expression, single-exon, employing a read coverage threshold, unreliable fragments, and annotated non-lncRNA transcripts (e.g., annotated protein-coding genes, pre-microRNAs, tRNAs, rRNAs, and pseudogenes) and transcripts with coding potential were eliminated. Finally, 4685, 3578, and 3487 rigorous lncRNA transcripts were, respectively, obtained from the UDCs, ADCs, and MDCs.

Table 1 shows the top 20 lncRNAs that are expressed in the three different cells based on RPKM. Similar to housekeeping genes, some lncRNAs are highly expressed in all the three different cells, which include ENSMUST00000192833, ENSMUST00000182520, and ENSMUST00000174808. However, several lncRNAs exhibited cell type-specific expression. For example, ENSMUST00000196219 and ENSMUST00000180396 were only highly expressed in the ADCs.

### 2.3. DE lncRNAs

The heatmap of DE lncRNAs showed that the expression profiles of lncRNAs in the three cell types exhibited extreme significant differences (Figure 3A). KCL cluster analysis indicated that the expression profiles of lncRNAs were more similar in the UDCs and MDCs, whereas the ADCs showed a distinct gene expression pattern.

Huge differences in lncRNA expressions were observed between ADCs and MDCs (Figure 3B–D). Approximately 711 lncRNAs were specifically expressed in MDCs and 803 lncRNAs were only expressed in ADCs. However, most of the cell-specific lncRNAs exhibited very low expression (RPKM < 1). We further focused on the DE lncRNAs between the MDCs and ADCs. A total of 836 lncRNA showed significant differences in expression levels between ADCs and MDCs (*t*-test, FDR < 0.05). In Figure 3E, except for the low-expression genes (RPKM < 100), there were still 43 lncRNAs that were significantly upregulated and 71 lncRNA transcripts were significantly downregulated in ADCs (FDR < 0.001 and fold-change > 2). These 114 significant DE lncRNAs were considered to be core lncRNAs, which possibly function as regulators of adipogenesis of myoblasts. Figure 3F show the top 10 upregulated and 10 downregulated lncRNA transcripts.

Moreover, a total of 1065 protein-coding genes showed altered expression levels in ADCs relative to MDCs (FDR < 0.01), of which 171 mRNAs were significant differential expression, with FDR < 0.001 and fold change > 2 (Appendix A). Nine mRNAs transcripts (ENSMUST00000023934, ENSMUST00000098192, ENSMUST00000153218, ENSMUST00000026907, ENSMUST00000112832, ENSMUST00000040153, ENSMUST00000165430, ENSMUST00000144897, and ENSMUST00000037023) showed an extreme increase in ADCs (log2 FC > 5), which indicated that these mRNAs control or affect lipogenesis in muscle cells. Table 2 shows the top 30 mRNA transcripts exhibiting significant changes in expression.

KEGG pathway analysis indicated that the 171 altered mRNA transcripts enriched several pathways that were related to nutrient metabolism in cells such as tyrosine (*P* = 0.035), glycolysis (*P* = 0.049), and fructose metabolism (*P* = 0.098) (Appendix A). These findings suggest that the distinct differentiation of cells determine the variations in intracellular nutrient metabolism and utilization.

### 2.4. QRT-PCR Validation

Next, to validate our RNA-seq results, we selected three upregulated and three downregulated significant DE lncRNAs for qRT-PCR assay. Figure 4 shows that the qPCR data on the expression of selected lncRNAs coincided with our sequencing data, thereby indicating that our transcript identification and abundance estimation are highly reliable.

### 2.5. Functional Prediction of DE lncRNAs

We predicted the potential targets of these lncRNAs in *cis*- and *trans*- regulation to determine the functions of the 114 core DE lncRNAs. For the *cis*- regulation of lncRNAs, we searched for DE mRNAs that were situated 100 kb upstream and downstream of the lncRNAs. The RIsearch software was used to analyze the *trans*-regulated mRNAs, and the threshold of free energy was <−100. Finally, 728 *trans* target genes and 118 *cis* targets for 77 core lncRNAs were obtained.

The functions of these target genes were annotated by GO and KEGG pathway enrichment analyses (Figure 5). A total of 255 GO entries with *P* < 0.05 were obtained, and Figure 5A shows the top 30 GO entries with *P* < 0.01. Moreover, several of the GO items with *P* < 0.05 were related to muscle cell differentiation and muscle formation, including GO:0048747 (muscle fiber development), GO:0051145 (smooth muscle cell differentiation), GO:0016203 (muscle attachment), and GO:0042692 (muscle cell differentiation).

Figure 5B shows the results of enrichment analysis of KEGG signaling pathways. The target genes were involved in some of the signaling pathways responsible for lipid metabolism and energy homeostasis with *P* < 0.1, such as the PPAR and mTOR pathways. In addition, some of the genes were involved in the MAPK (*P* = 0.3) and the JAK-STAT (*P* = 0.17) pathways, which are closely related to lipogenesis. These results suggest that DE lncRNAs and their target genes regulate the adipogenic transdifferentiation of muscle cells by regulating cell fate determination, intracellular gene reprogramming, cell differentiation, and substance metabolism.

In addition, several targets enriched the functional category of mesodermal cell fate determination (GO:0007500) (Figure 6). Both muscle and adipose cells originated from the common mesoderm cells in mammals, which thereby suggested that the corresponding lncRNAs could mediate the initial determination of cellular fate. Eight DE lncRNAs may thus possibly influence mesodermal cell fate by targeting BMP4, a member of the BMP family, which controls the differentiation of myoblasts as well as regulates adipogenesis in adipocytes [10,29,30]. The DGE-seq data also showed that BMP4 is upregulated in ADCs.

### 2.6. lncRNA-GM43652 Plays a Role in the Adipogenesis of C2C12 Cells

In the present study, ENSMUST00000196219 (name: *lncRNA-Gm43652*) showed the highest expression level and the largest fold change in expression in ADCs as compared to MDCs. Therefore, we inferred that this gene might be directly related to cell adipogenesis. Additionally, the expression level of ENSMUST00000196219 at different time points during the adipogenesis of C2C12 cells was determined by qRT-PCR, and the results indicated that the expression of the *lncRNA-Gm43652* gene decreased during the early stages of transdifferentiation and was later consistently upregulated (Figure 7A). Furthermore, knocking down *lncRNA-Gm43652* by siRNA transfection suppressed lipid deposition in cells. Figure 6 shows that the expression of *lncRNA-Gm43652* in cells decreased by 73% and 57% 24 h and 72 h after siRNA transfection as compared to the control cells (Figure 7B). Intracellular lipid deposition also decreased by approximately 30% (based on Oil Red O staining), and FABP4 and FAS were downregulated after knocking down *lncRNA-Gm43652*. Taken together, these findings indicate that *LncRNA-Gm43652* is a candidate regulator of adipogenic transdifferentiation of myoblasts.

## 3. Discussion

The main function of skeletal muscle is to contract to facilitate movement of the body and to provide energy storage and expenditure. In addition, skeletal muscle is the largest endocrine organ in humans and animals that produces and releases many cytokines. These proteins that are released by skeletal muscle tissues could influence metabolism in other organs, such as adipose and liver [31]. Therefore, normal growth and the stability functions of skeletal muscle may influence the body’s metabolic balance and energy homeostasis. Additionally, the growth and development of skeletal muscle also determines the meat production of livestock and the meat quality we can obtain [32].

lncRNAs have emerged as a class of important molecules that regulates gene expression at the epigenetic, transcriptional, and post-transcriptional levels through a wide array of mechanisms [33]. Numerous lncRNAs that are related to growth and development of muscle in humans and animals have been identified through sequencing and microarray projects. These lncRNAs participate in all of the processes of skeletal muscle development, including proliferation, differentiation, and fusion of myocytes, muscle hypertrophy, and conversion of muscle fiber types by targeting pro- or anti-myogenic genes. For example, lnc-MD1 could stimulate myogenesis and muscle cell differentiation by upregulating the expression of MEF2C and controlling transcriptional co-activator 1 by competitive binding with microRNAs, such as miR-135 and miR-133 [21].

The adipogenic conversion of muscle cells may change cellular fate when numerous genes are reprogrammed. Molecular regulation and metabolic physiology may change to address the requirements of adipogenesis during transdifferentiation. In recent years, we have gained new insights into the adipogenesis in muscle cells. To our knowledge, the formation of muscle, bone, and adipose tissues involves a multistep processes that includes the determination of a common progenitor mesodermal cell toward a specific differentiation pathway, followed by the expression of various terminal differentiation phenotypes [34]. In vitro studies have demonstrated the multi-directional differentiation potential of muscle-derived stem cells or precursor cells [1,2,3,4,5,6,35,36]. Our previous study has shown that some miRNAs participate in the control of adipogenesis in muscle cells, such as miR-199a, which negatively regulates the transdifferentiation of C2C12 myoblasts by targeting the *FATP1* gene [15]. In this study, our analysis has also shown that several lncRNA genes are involved in the determination of cellular fate by affecting BMP4, which is a key molecular switch for myogenesis and adipogenesis. Additionally, KEGG pathway analysis of coding genes in the present study has revealed that the absorption and utilization of nutrients in the adipogenic and myogenic C2C12 cells are different.

In addition, some reports have indicated that the fat cells transformed from muscle precursor cells exhibit the characteristics of brown adipocytes. Lineage-tracing experiments have shown that brown adipocytes, skeletal muscle cells, and dorsal dermal cells are all derived from the same multi-potential progenitor cells that originate from the central dermomyotome [37]. A study has revealed that Myf5-expressing progenitors can give rise to both skeletal muscle and brown fat cells [38]. The transcriptional regulator PRDM16 controls the switch of cell fate [39,40]. An et al. have also shown that the MyoD/Myf5-E2F4/p107/p130 axis functions as a molecular switch in postnatal myoblasts, which regulates the choice between differentiation into myoblasts and brown adipocytes [41]. We have also observed differences in lipid droplet morphology in adipogenic C2C12 cells compared to 3T3-L1 adipocytes using the similar induction conditions. UCP1 and PRDM16, two markers for brown adipocytes, have shown higher expression levels in adipogenic C2C12 cells than 3T3-L1 adipocytes, indicating that adipogenic myoblasts are more similar to brown adipocytes.

Although current knowledge on the mechanism of adipogenesis conversion of muscle cells is expanding, our understanding of the changes and function of lncRNAs in the transformation process remains limited. The present study has revealed huge differences in the gene expression profiles between normal myogenic differentiation and adipogenic transdifferentiation of myoblasts while using high-throughput RNA sequencing. A total of 1065 DE mRNA transcripts and 836 DE lncRNA transcripts between adipogenic cells and myogenic cells were identified in this study. Approximately 114 lncRNAs and 171 mRNAs were selected as core candidate regulators that are involved in the regulation of muscle cell transdifferentiation and subjected to further investigation.

We further predicted the target mRNAs for core lncRNAs while using *cis* and *trans* regulation analysis, and the results showed that there were 846 mRNAs for 77 lncRNAs that participate in the control of cell fate determination, intracellular gene reprogramming, cell differentiation, and substance metabolism. However, the biological functions of lncRNAs are highly diverse, and lncRNAs generally exert their effects on cells through different mechanisms. A number of studies have shown that lncRNAs can serve as molecular guides for chromatin-modifying complexes to their target genes, act as molecular signals in response to DNA damage, function as protein or miRNA decoys to regulate mRNA expression, and prevent binding to their targets or elements [42]. The sequencing results of the present study have revealed that several lncRNAs may be related to the adipogenic conversion of muscle cells. However, that is not sufficient for fully understanding the regulatory functions of lncRNAs in transformation.

The in-depth study and exploration on the molecular mechanism of adipogenesis in muscle cells will improve our understanding of the process of transformation between muscle and adipose, the metabolic functions of muscle tissues, and the pathogenesis of muscular diseases. Our present study utilized RNA sequencing to provide useful information regarding the changes in lncRNA expression during the adipogenesis of myoblasts. Furthermore, we have generated a list of candidate lncRNAs that are related to myoblast transdifferentiation that may be employed in future research studies.

## 4. Materials and Methods

### 4.1. Cell Culture

Mouse C2C12 myoblasts were resuscitated and cultured in Dulbecco’s modified Eagle’s medium (DMEM)-F12 medium (containing 10% fetal calf serum and 100 U/mL penicillin/streptomycin; GIBCO, Grand Island, NY, USA) and then passaged when the confluency reached 90%. The passaged cells were plated into six-well culture plates and then induced to undergo myogenic differentiation and adipogenic transdifferentiation. The myogenic differentiation was induced in subconfluent cells in differentiation medium (DMEM, 2% horse serum, 100 U/mL penicillin/streptomycin). The adipogenic trans-differentiation was inducted in adipogenesis medium (DMEM-F12, 20% fetal calf serum, 10 μg/mL insulin, 1 μM dexamethasone, 0.5 mM 3-isobutyl-1-methylxanthine, and 10 μM rosiglitazone, Sigma–Aldrich, St. Louis, MO, USA). The Giemsa staining and Oil red O staining (Solarbio, Shanghai, China) were used to determine the myogenesis and adipogenesis in different differentiated cells respectively as our previous describe [43]. All of the experiments were performed while using three separate cell clones.

### 4.2. Oil Red O and Giemsa Staining

The status of adipogenic transdifferentiation and adipogenesis in C2C12 cells were assessed by the Oil Red O method. Briefly, the cells were washed three times with phosphate buffered saline (PBS, pH 7.2) and fixed in 4% paraformaldehyde for 30 min at ambient temperature. The fixed cells were washed three times in PBS again and then incubated in staining solution (60% Oil Red O stock solution and 40% H_2_O) for 30 min at ambient temperature. The cells were washed twice with deionized water and observed under an inverted microscope.

During myogenic differentiation, the multinucleated myotubes were visualized in C2C12 cell cultures with the Giemsa staining. Cells were washed twice with PBS, fixed in 4% paraformaldehyde for 30 min, and then stained for 1 min with Giemsa stain solution (Solarbio, Beijing, China) at ambient temperature. Cells were washed twice with deionized water and observed under an inverted microscope.

### 4.3. Sample Collection

Undifferentiated cells (UDCs), cells that underwent myogenesis induction for five days (myogenic differentiated cells, MDCs), and cells subjected to adipogensis induction for eight days (adipogenic differentiated cells, ADCs) were separately collected for total RNA extraction. Three cell clones per treatment were pooled into one sequencing sample.

### 4.4. DGE-seq Sequencing and Data Processing

Total RNA was extracted using TRIzol reagent (Invitrogen, CA, USA) while following the manufacturer’s procedure. The total RNA quality and quantity were analysis of Bioanalyzer 2100 and RNA 6000 Nano Lab Chip Kit (Agilent, CA, USA) with RIN number >7.0. Approximately 10 μg of total RNA was used to remove the ribosomal RNA according to the manuscript of the Epicentre Ribo-Zero Gold Kit (Illumina, San Diego, CA, USA). Following purification, the ribo-minus RNA fractions is fragmented into small pieces using divalent cations under elevated temperature. Subsequently, the cleaved RNA fragments were reverse-transcribed to create the final cDNA library in accordance with a strand-specific library preparation by dUTP method. Afterwards, we performed the single end sequencing on an Illumina HiSeq 2000/2500 at the LC Sciences (Hangzhou, China), following the vendor’s recommended protocol.

The raw data containing adaptor sequences, tags with low quality sequences and unknown nucleotides N were filtered to obtain clean reads of 36 nt length. The clean reads were then subjected to quality assessment. These included classification of total and distinct reads and assessment of their percentage in the library, analysis of saturation of the library, and correlation analysis of biological replicates. Bowtie mapped all clean tags to the transcripts sequence; only 1-bp mismatch is allowed. For monitoring the mapping events on both strands, both the sense and the complementary antisense sequences were included in the data collection. The number of perfect clean reads corresponding to each gene was calculated and normalized to the number of reads per kilobase of exon model per million mapped reads (RPKM).

### 4.5. Analyses of Differential Expressed (DE) lncRNAs

Based on the expression levels, significant DE lncRNAs among different cell samples were identified with *P*-value ≤ 0.05 and log2 fold-change (log2 FC) ≥ 1. The cluster of the DEGs was performed while using common Perl and R scripts.

### 4.6. Target Genes of DE lncRNAs and Enrichment Analysis

The regulation of lncRNAs on target genes is mainly divided into two categories, namely cis and trans. The *cis*-regulatory targets of lncRNAs are predicted based on their positional relationships, defining DE lncRNAs and DE mRNAs within each 100-kb region of chromosomes. The target genes of lncRNAs exhibiting *trans* regulation are predicted mainly based on the free energy that is needed to form secondary structures between lncRNAs and mRNA sequences. In this study, the RNAplex (http://www.tbi.univie.ac.at/software/) algorithm was used to predict the trans targets of DE lncRNAs. Finally, all the *cis*- and *trans*-regulated genes were taken as the targets for the DE lncRNAs. Gene Ontology (GO) functional analysis was used to determine the main biological functions of the candidate target genes. KEGG metabolic pathway analysis identified the most important biochemical and signal transduction pathways that were involved in the candidate target genes.

### 4.7. qRT-PCR

Six DE lncRNAs were selected for qRT-PCR assay to verify the DGE-Seq results. In addition, the mRNA levels of several key adipogenic genes or myogenic genes were detected by qRT-PCR. The PCR detection was performed while using the Q6 qPCR system with SYBR Premix Ex Taq II (TaKaRa) and normalized using GAPDH as the endogenous control. All of the reactions were prepared while using three replicates and the expression levels of genes were expressed as fold change using the 2^−ΔΔCT^ method.

### 4.8. SiRNA Transient Knockdown

A siRNA of lncRNA-GM43652 and a nonspecific duplex (negative control) were custom synthesized by Biotech Co. (Nantong, China). The siRNA transfected into the adipogenic C2C12 cells after four days of induction using Rfect siRNA Transfection Reagent (BIO-TRAN), according to the manufacturer’s protocol. The cells were performed Oil red O staining, BODIPY staining, and qPCR analysis four days after transfection.

### 4.9. Statistical Analysis

The data were processed while using R, Microsoft Excel, and GraphPad Prism 5.0. Differences among groups were analyzed using one-way analyses of variance (ANOVAs), followed by Dunnett’s test. Two-tailed student’s *t*-test was used for comparison between two groups, and *P* < 0.05 was considered to be statistically significant.

## Figures and Tables

**Figure 1 ijms-20-03725-f001:**
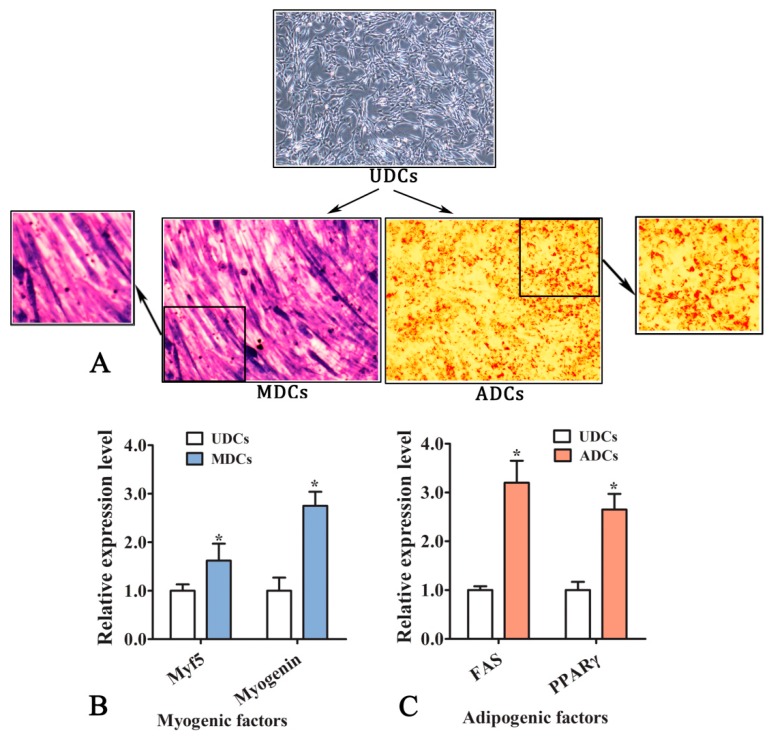
The myogenic differentiation and the adipogenic transdifferentiation of C2C12 myoblasts. (**A**) Morphological difference between adipogenic differentiated cells (ADCs) and myogenic differentiated cells (MDCs). MDCs are stained with Giemsa after six days of the myogenesis differentiation induction, and ADCs are stained with Oil red O after eight days of the adipogenesis differentiation induction. UDCs, undifferentiated cells. (**B**) Relative mRNA levels of the myogenic genes in MDCs and UDCs. (**C**) Relative mRNA levels of the adipogenic genes in ADCs and UDCs. The expression levels were detected by qRT-PCR, *n* = 5. The data are presented as the means ± SEM. * *P* ≤ 0.05, student’s *t*-test.

**Figure 2 ijms-20-03725-f002:**
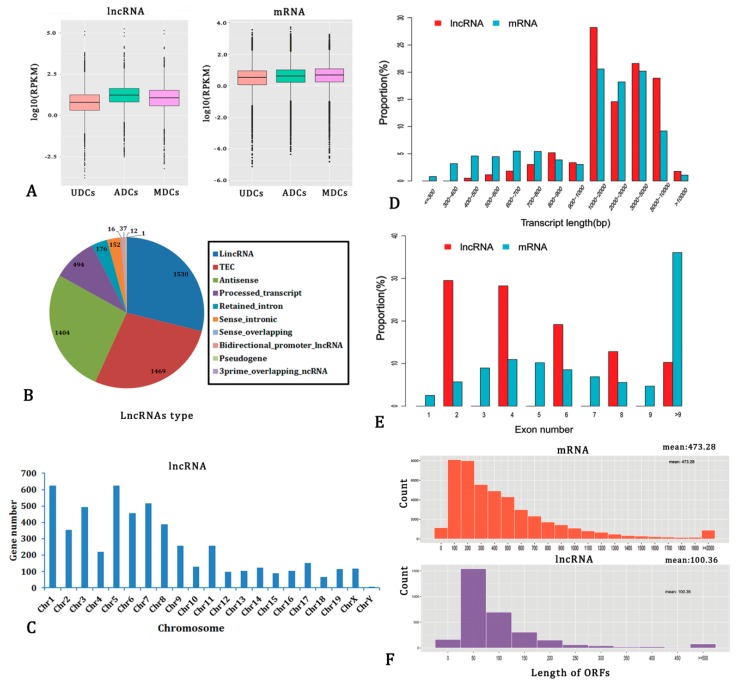
The features of long non-coding RNA (lncRNAs) and mRNAs. (**A**) The FPKM distribution of all identified lncRNAs and mRNAs at three different cells. (**B**) Percentage of different types of lncRNAs. (**C**) Distribution of lncRNA transcripts along each chromosome. (**D**) Transcript size distribution of lncRNAs and mRNAs. (**E**) Exon number per transcript of lncRNAs and mRNAs. (**F**) Distribution of the open reading frame (ORF) length of lncRNAs and mRNAs.

**Figure 3 ijms-20-03725-f003:**
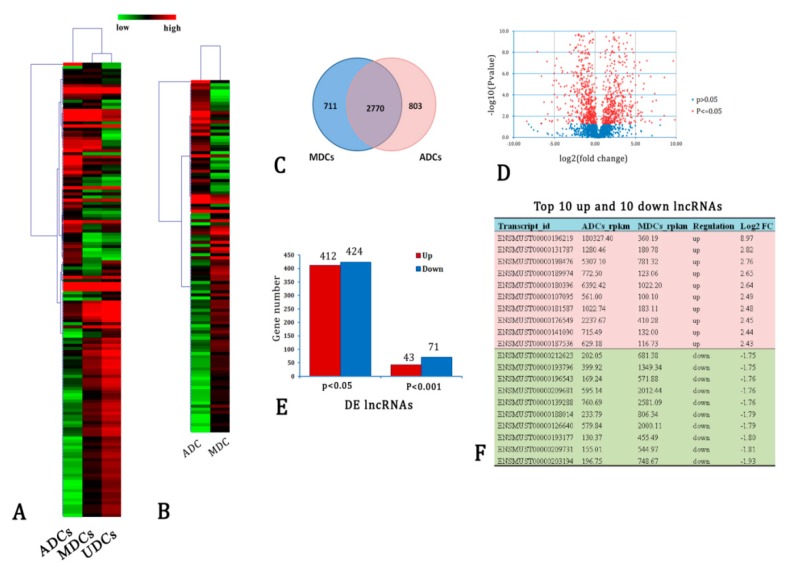
Differentially expressed (DE) lncRNA transcripts in adipogenic differentiated cells (ADCs)/myogenic differentiated cells (MDCs). (**A**) Heatmap showing that ADCs exhibit a distinct lncRNA expression profile relative to MDCs and UDCs. Red indicates high expression levels, and green represents low expression levels. (**B**) Heatmap of DE lncRNAs in ADCs/MDCs. (**C**) Venn diagram of all expressed lncRNAs in ADCs/MDCs. (**D**) Volcano plot of the DE lncRNAs. Red indicates *P* ≤ 0.05. (**E**) Number of DE lncRNAs with *P* ≤ 0.05 and *P* ≤ 0.001. (**F**) Top 10 upregulated and top 10 downregulated lncRNAs in ADCs/MDCs.

**Figure 4 ijms-20-03725-f004:**
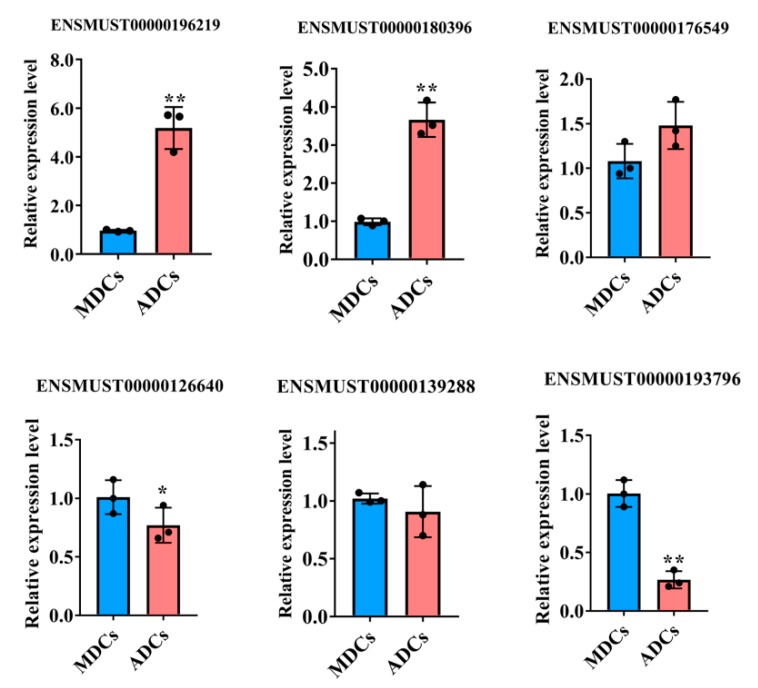
qPCR validation of the RNA-Sequencing (RNA-Seq) expression results of lncRNAs (six differentially expressed genes). The data are presented as the means ± SEM. *n* = 3, ** *P* ≤ 0.01, * *P* ≤ 0.05, Student’s *t*-test.

**Figure 5 ijms-20-03725-f005:**
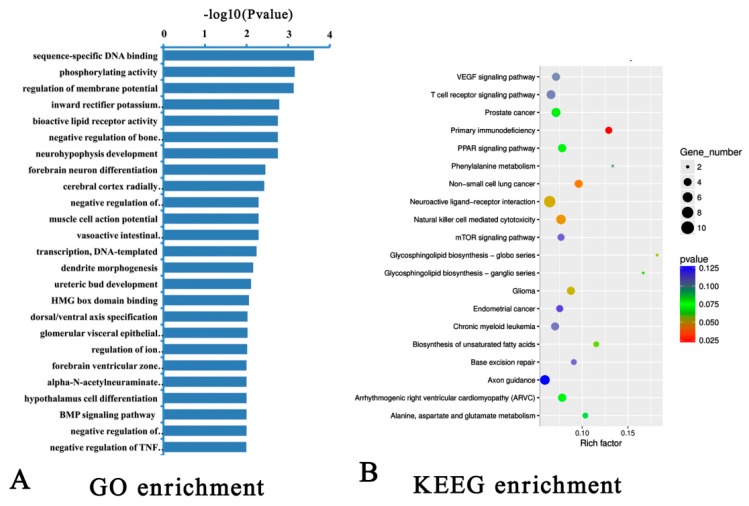
The top Gene Ontology (GO, (**A**)) and Kyoto Encyclopedia of Genes and Genomes (KEGG, (**B**)) enrichment analyses of the targets of differentially expressed lncRNAs.

**Figure 6 ijms-20-03725-f006:**
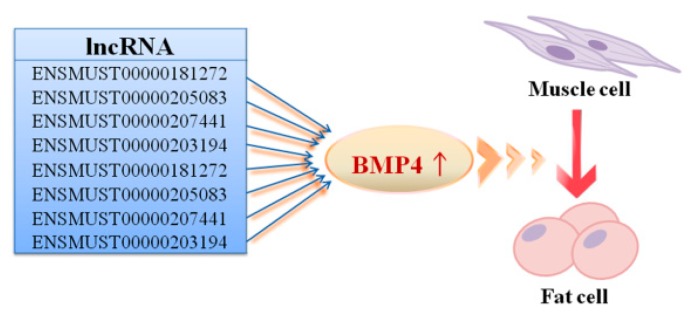
The target predication and GO analysis indicate that eight lncRNAs are related to cell fate decision in myoblasts based on the target gene, BMP4.

**Figure 7 ijms-20-03725-f007:**
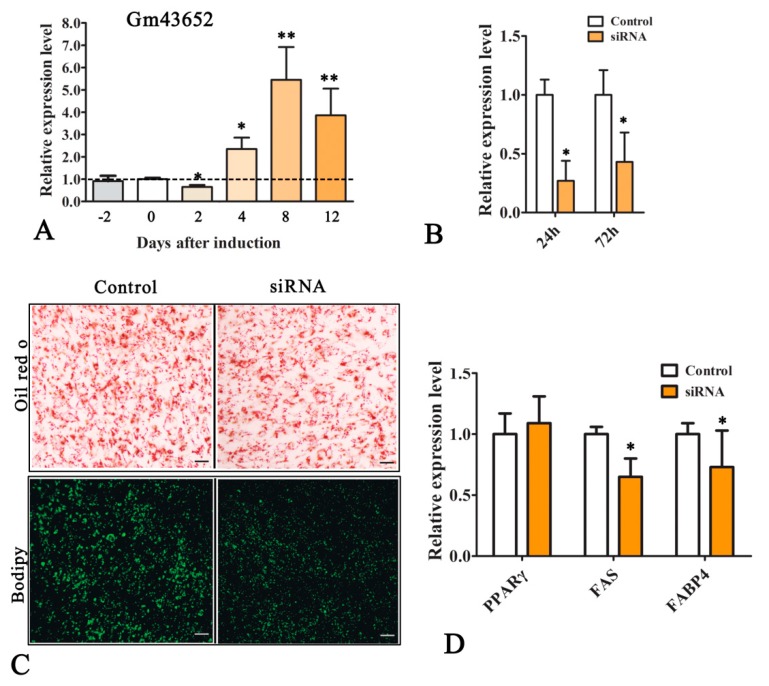
The *lncRNA-GM43652* gene is a potential regulator of the adipogenic conversion of C2C12 myoblasts. (**A**) Changes in the expression of *lncRNA-GM43652* during adipogenic transdifferentiation of myoblasts. (**B**) Knock down of *lncRNA-GM43652* using small interfering RNAs (siRNAs). (**C**) Knocking down *lncRNA-GM43652* decreased lipid deposition in the ADCs. The lipids in cells are stained with Oil Red O and BODIPY. (**D**) Knocking down *lncRNA-GM43652* decreased the mRNA levels of the adipogenic genes. Expression levels were detected by qRT-PCR, *n* = 3. The data are presented as the mean ± SEM. * *P* ≤ 0.05, student’s *t*-test.

**Table 1 ijms-20-03725-t001:** The top expressed 20 lncRNA transcripts in the UDCs, MDCs, and ADCs.

UDCs		ADCs		MDCs	
lncRNAs	RPKM	lncRNAs	RPKM	lncRNAs	RPKM
ENSMUST00000192833	124,640.19	ENSMUST00000196219	180,327.40	ENSMUST00000192833	145,216.23
ENSMUST00000182520	77,450.29	ENSMUST00000192833	104,112.48	ENSMUST00000182520	90,764.06
ENSMUST00000174808	6608.49	ENSMUST00000182520	65,232.66	ENSMUST00000174808	14,136.14
ENSMUST00000139288	5952.70	ENSMUST00000172812	15,463.37	ENSMUST00000173314	12,237.22
ENSMUST00000172812	5176.44	ENSMUST00000174808	13,599.96	ENSMUST00000172812	11,376.69
ENSMUST00000173314	5173.88	ENSMUST00000173314	12,106.39	ENSMUST00000181751	3401.46
ENSMUST00000209681	4609.90	ENSMUST00000173499	6486.05	ENSMUST00000174784	3368.69
ENSMUST00000126640	4586.59	ENSMUST00000148202	6416.90	ENSMUST00000173523	3034.68
ENSMUST00000123278	4550.22	ENSMUST00000180396	6392.42	ENSMUST00000173499	2907.27
ENSMUST00000181751	4292.44	ENSMUST00000173523	5893.02	ENSMUST00000139288	2581.09
ENSMUST00000187415	4249.52	ENSMUST00000198476	5307.10	ENSMUST00000192176	2570.33
ENSMUST00000118575	3575.11	ENSMUST00000174784	5116.32	ENSMUST00000140716	2219.49
ENSMUST00000148202	3258.77	ENSMUST00000209541	4304.59	ENSMUST00000148202	2035.23
ENSMUST00000193796	3055.50	ENSMUST00000122365	4108.33	ENSMUST00000123278	2018.85
ENSMUST00000192629	2613.42	ENSMUST00000192176	3958.38	ENSMUST00000209681	2012.44
ENSMUST00000211359	2580.81	ENSMUST00000181751	3701.19	ENSMUST00000126640	2000.11
ENSMUST00000146654	2203.23	ENSMUST00000182010	2920.89	ENSMUST00000187415	1903.12
ENSMUST00000134427	2191.40	ENSMUST00000187351	2389.74	ENSMUST00000182010	1777.71
ENSMUST00000181631	2063.67	ENSMUST00000072769	2353.86	ENSMUST00000152754	1735.28
ENSMUST00000192994	2042.13	ENSMUST00000176549	2237.67	ENSMUST00000136359	1727.49

**Table 2 ijms-20-03725-t002:** The top 30 changed mRNA transcripts in ADCs vs. MDCs.

Accession	ADCs_rpkm	MDCs_rpkm	Regulation	log2 FC	FDR	Gene Symbol
ENSMUST00000000466	503.48	89.23	up	2.50	0.0000	Plin2
ENSMUST00000000756	384.21	96.18	up	2.00	0.0000	Rpl13
ENSMUST00000004072	1336.05	278.51	up	2.26	0.0000	Rpl8
ENSMUST00000008036	681.50	233.72	up	1.54	0.0000	Rplp1
ENSMUST00000008812	826.36	176.26	up	2.23	0.0000	Rps18
ENSMUST00000009039	512.00	198.39	up	1.37	0.0000	Rpl30
ENSMUST00000017548	856.60	192.56	up	2.15	0.0000	Rpl19
ENSMUST00000017610	168.79	463.18	down	−1.46	0.0000	Timp2
ENSMUST00000018437	459.23	126.45	up	1.86	0.0000	Pfn1
ENSMUST00000020238	953.64	279.12	up	1.77	0.0000	Hsp90b1
ENSMUST00000020640	572.53	199.28	up	1.52	0.0000	Rack1
ENSMUST00000020909	155.75	471.23	down	−1.60	0.0000	Laptm4a
ENSMUST00000021822	27.18	341.94	down	−3.65	0.0000	Ogn
ENSMUST00000021933	576.35	233.46	up	1.30	0.0000	Ctsl
ENSMUST00000022704	251.33	644.66	down	−1.36	0.0000	Itm2b
ENSMUST00000023269	553.91	167.99	up	1.72	0.0000	Rpl24
ENSMUST00000023934	1074.34	0	up	Inf	0.0000	Hbb-bs
ENSMUST00000025052	350.69	68.09	up	2.36	0.0000	Rps10
ENSMUST00000025511	339.15	36.65	up	3.21	0.0000	Rps14
ENSMUST00000025563	2703.24	1266.10	up	1.09	0.0000	Fth1
ENSMUST00000026565	148.88	364.95	down	−1.29	0.0000	Ifitm3
ENSMUST00000026907	1200.90	2.96	up	8.66	0.0000	Klkb1
ENSMUST00000027409	521.98	232.54	up	1.17	0.0000	Des
ENSMUST00000028222	364.77	99.43	up	1.88	0.0000	Hspa5
ENSMUST00000029722	1196.45	427.35	up	1.49	0.0000	Rps3a1
ENSMUST00000031318	17.93	417.80	down	−4.54	0.0000	Cxcl5
ENSMUST00000031617	350.05	135.61	up	1.37	0.0000	Rpl6
ENSMUST00000032934	356.74	154.31	up	1.21	0.0000	Aldoa
ENSMUST00000033683	577.73	200.73	up	1.53	0.0000	Rps4x
ENSMUST00000033741	198.11	785.72	down	−1.99	0.0000	Bgn

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
