# Peer review of "Comparison of LncRNA Expression Profiles during Myogenic Differentiation and Adipogenic Transdifferentiation of Myoblasts"

_ijms, 2019, doi:10.3390/ijms20153725_

Reviewer 1 Report

The authors showed a possible molecular pathway involved in the myogenic to adipogenic trans differentiation. The results could be improved adding a dot plots graphs instead of bar graph for the qPCR data and time lapse microscopy for the trans differentiation results (or pictures with high resolution).

Author Response

We thank the reviewer for this suggestion; we have changed the qPCR data in Figure 4. In addition, representative and recognizable images are now used in the paper. The details of cell morphological changes are highlighted in Figure 1. However, we cannot provide cell images at a higher resolution because we do not have a laser confocal microscope or other advanced imaging equipment available to us.

Reviewer 2 Report

To investigate the expression and function of lncRNAs in the adipogenic transdifferentiation of muscle cells, the authors identified 114 core lncRNAs expressed in adipogenic cells and predicted the potential targets of these lncRNA, Furthermore, the authors showed that lncRNA-GM43652 regulates adipogenic transdifferentiation of C2C12 myoblasts. This manuscript provides useful information for the clarification of molecular mechanism of muscle-adipose conversion. However, to improve this manuscript, the authors should address the following points.

Major comments:

1. The authors identified total 1065 protein coding genes expressed in ADCs (supplementary Figure 1) and described that 171 altered mRNA transcripts were enriched several pathway. However, only top 30 genes were shown in Table 2. The authors should show all of the altered mRNA transcripts in supplementary table.

2. The authors predicted the potential targets (728 trans target and 118 cis target genes) of 77 core DE lncRNAs. The authors should show all of the targets in supplementary table.

3. The authors speculated the 8 DE lncRNAs influence mesodermal cell fate by targeting BMP4. The authors should investigate whether the inhibition of some lncRNAs in C2C12 cells affects the expression levels of BMP4.

Minor comments:

1. Is the data in Figure 2B-F obtained from UDCs, ADCs, or MDCs?

2. In page 5, line 161, is “Top 20” “Top 30”?

Author Response

Major comments:

1. The authors identified total 1065 protein coding genes expressed in ADCs (supplementary Figure 1) and described that 171 altered mRNA transcripts were enriched several pathway. However, only top 30 genes were shown in Table 2. The authors should show all of the altered mRNA transcripts in supplementary table.

R: Many thanks for your good suggestions. All 117 significant altered mRNAs have been listed in the supplementary table 1.

2. The authors predicted the potential targets (728 trans target and 118 cis target genes) of 77 core DE lncRNAs. The authors should show all of the targets in supplementary table.

R: All cis-regulated targets have been listed in the supplementary table 2 and All trans-regulated targets have been listed in the supplementary table 3.

3. The authors speculated the 8 DE lncRNAs influence mesodermal cell fate by targeting BMP4. The authors should investigate whether the inhibition of some lncRNAs in C2C12 cells affects the expression levels of BMP4.

R: Our in silico analysis suggested eight DE lncRNAs may mediate the change in cellular fate by affecting BMP4, a potential switch factor for myogenesis and adipogenesis. However, this is a relatively huge research topic and requires further investigation in the near future.

 Minor comments:

1. Is the data in Figure 2B-F obtained from UDCs, ADCs, or MDCs?

R: Yes, the count data obtained from the three cell types.

2. In page 5, line 161, is “Top 20” “Top 30”?

R: This mistake has been corrected in the revised manuscript. 

Reviewer 3 Report

In this manuscript, Qi et al. described the potential role of long-chain non-coding RNAs (lncRNAs) in the process of adipogenic transdifferentiation of myoblasts. The authors used RNA sequencing to identify differentially expressed lncRNAs and mRNAs in myoblasts under distinct differentiation conditions. They further showed that one of highly expressed lncRNAs, lncRNA-GM43652, may regulate the transdifferentiation process. While this study unravels a new unidentified role of lncRNAs, a few critical concerns need to be addressed before it is in a publishable fashion. Specific comments are as follows:

1) The whole project was based on the results using a well characterized cell line, C2C12. The authors should try to validate or discuss whether these lncRNA candidates can represent or are also crucial in transdifferentiation of primary cells derived from mice or even humans, e.g. test the expression levels or their adipogenic potential.

2) In Supplementary Figure 1A, the authors compared the number/size and gene expression between adipogenic myoblasts and 3T3-L1 cells. In line 91, it says "at the same differentiation induction time". In the figure legends, its says "under same induction conditions". As the commonly used 3T3-L1 differentiation protocol is different from what is described in the Materials and Methods (section 4.1) for myoblasts, the authors need to clarify the methods for these comparisons.

3) As described in the methods section 4.3, C2C12 cells were induced for myogenesis for 5 days and adipogenesis for 8 days. Did quantitative RT-PCR validation use the same culture conditions? How many times were these experiments performed to have results shown in Figure 4?

4) In Figure 6, 8 lncRNAs were predicted to be able to up-regulate BMP4 expression. Can this be experimentally approved? The authors also described that BMP4 levels were higher in ADCs than those in MDCs (line 217). Once again, can this be experimentally validated?

5) In Figure 7, lncRNA-GM43652 was knocked down before induction for adipogenesis. Is there a particular reason to examine lipid deposition and gene expression on day 4 instead of day 8 as in other experiments?

6) As in 5), in Figure 6C, the authors described that "intracellular lipid deposition was also decreased by approximately 30%". How was this determined? In Figure 6D, is there a significant difference between the control and siRNA group in FABP4 expression? Is there a possible explanation that there is no difference in PPARgamma expression but there is in its target gene fatty acid synthase?

7) Is the expression change in lncRNA-GM43652 specific for adipogenic transdifferentiation in myoblasts or adipogenic differentiation in general? Does it function in e.g. 3T3-L1 differentiation?

 Minor concers:

1) A few typing and grammar errors need to be corrected. For example, fructose metabolism (line 164); protein decoys(?) (line 295), and 3T3-L1 (Supplementary Figure 1A legends). The sentence "The DGE-seq data also showed that BMP4 increases the expression in ADCs compared that in MDCs." (line 216) does not make sense.

2) In Supplementary Figure 2A, what do the 9 genes in the box stand for?

Author Response

1. The whole project was based on the results using a well characterized cell line, C2C12. The authors should try to validate or discuss whether these lncRNA candidates can represent or are also crucial in transdifferentiation of primary cells derived from mice or even humans, e.g. test the expression levels or their adipogenic potential.

R: Thank you for your suggestions. In this study, all our findings and conclusions were obtained using the C2C12 cell line. We will also continue to evaluate the expression and function of lncRNAs and other non-coding genes on the primary muscle cells in our future studies. We are currently working on elucidating the adipogenesis mechanism of skeletal muscle satellite cells of pigs.

 2. In Supplementary Figure 1A, the authors compared the number/size and gene expression between adipogenic myoblasts and 3T3-L1 cells. In line 91, it says "at the same differentiation induction time". In the figure legends, its says "under same induction conditions". As the commonly used 3T3-L1 differentiation protocol is different from what is described in the Materials and Methods (section 4.1) for myoblasts, the authors need to clarify the methods for these comparisons.

R: The right description should be "at the same differentiation induction time" in the Supplementary Figure 1 legend.

3. As described in the methods section 4.3, C2C12 cells were induced for myogenesis for 5 days and adipogenesis for 8 days. Did quantitative RT-PCR validation use the same culture conditions? How many times were these experiments performed to have results shown in Figure 4?

R: The same RNA samples were divided into two parts, namely, for RNA-sequencing and qRT-PCR detection. Three independent RNA samples per group were used for QPCR analysis.

4. In Figure 6, 8 lncRNAs were predicted to be able to up-regulate BMP4 expression. Can this be experimentally approved? The authors also described that BMP4 levels were higher in ADCs than those in MDCs (line 217). Once again, can this be experimentally validated?

 R: Our in silico analysis suggested eight DE lncRNAs may mediate changes in cellular fate by affecting BMP4, which is a potential switch factor for myogenesis and adipogenesis. However, this is a relatively hug research topic that requires further investigation in the near future.

 5. In Figure 7, lncRNA-GM43652 was knocked down before induction for adipogenesis. Is there a particular reason to examine lipid deposition and gene expression on day 4 instead of day 8 as in other experiments?

 R: We apologize for our vague description in the methodology section. In this study, siRNA was transfected into the C2C12 cells after four days of adipogenic induction, followed by lipid deposition and gene expression analyses (day 8 after induction).

 6. As in 5), in Figure 6C, the authors described that "intracellular lipid deposition was also decreased by approximately 30%". How was this determined? In Figure 6D, is there a significant difference between the control and siRNA group in FABP4 expression? Is there a possible explanation that there is no difference in PPARgamma expression but there is in its target gene fatty acid synthase?

R: Intracellular lipid deposition was assessed by Oil red O staining. After microscopic imaging, Oil red O dye in the cells was eluted with isopropanol, and the solution absorbance is detected with a spectrophotometer at a wavelength of 510 nm.

 Our qPCR results indicated that FABP4 and FAS were significantly downregulated with siRNA treatment, whereas PPARγ showed no significant change in expression. In general, PPARγ is upregulated during the early stage of adipogenic differentiation; however, FAS was continuously upregulated during the middle and late stages of differentiation. These results suggested that the observed difference in gene expressions may be caused by the differentiation stage and status of cells.

 7. Is the expression change in lncRNA-GM43652 specific for adipogenic transdifferentiation in myoblasts or adipogenic differentiation in general? Does it function in e.g. 3T3-L1 differentiation?

R: We observed that lncRNA-GM43652 expression in 3T3-L1 cells increases with cell differentiation. However, it is worth noting that the basal expression abundance of lncRNA-GM43652 at day 8 is significantly lower in 3T3-L1 cells (mean qPCR ct value: 27.83) compared to C2C12 cells (mean qPCR ct value: 16.47).

 Minor concerns:

1. A few typing and grammar errors need to be corrected. For example, fructose metabolism (line 164); protein decoys (?) (line 295), and 3T3-L1 (Supplementary Figure 1A legends). The sentence "The DGE-seq data also showed that BMP4 increases the expression in ADCs compared that in MDCs." (line 216) does not make sense.

R: These have been corrected in the revised manuscript.

 2. In Supplementary Figure 2A, what do the 9 genes in the box stand for?

R: The description of the nine DE mRNAs have been added to the revised manuscript.

Reviewer 4 Report

         The current study has identified novel candidate regulators that may be assessed in future molecular studies on adipogenic conversion of muscle cells.

        Skeletal muscle has recently been identified as an organ that produces and releases cytokines, which have been named “myokines”. Given that skeletal muscle is the largest organ in the human body, our discovery that contracting skeletal muscle secretes proteins sets a novel paradigm: skeletal muscle is an endocrine organ producing and releasing myokines in response to contraction which can influence metabolism in other tissues and organs [1].

         Long noncoding RNAs (lncRNAs) play a major role in adipogenesis. However, differential expression profiles of lncRNAs in inguinal white adipose tissue (iWAT) between wild-type (WT) and ob/ob mice, as well as their roles in adipogenesis, are not well understood [2]. A Novel lnc-RNA, Named lnc-ORA, Is Identified by RNA-Seq Analysis, and Its Knockdown Inhibits Adipogenesis by Regulating the PI3K/AKT/mTOR Signaling Pathway [2].

           Up-regulation of long noncoding RNA slincRAD expression was found to occur in the early differentiation stages of 3T3-L1 cell, prior to the regulation of major transcription factors [3]. Long non-coding RNA slincRAD functions in methylation regulation during the early stage of mouse adipogenesis [3].

  Authors are kindly requested to emphasize the current concepts about these issues in the context of recent knowledge and the available literature. This article should be quoted in the References list.

References

Have guidelines addressing      physical activity been established in nonalcoholic fatty liver disease?      World J Gastroenterol. 2012 Dec 14; 18 (46): 6790-800. doi:      10.3748/wjg.v18.i46.6790.

A Novel lnc-RNA, Named      lnc-ORA, Is Identified by RNA-Seq Analysis, and Its Knockdown Inhibits      Adipogenesis by Regulating the PI3K/AKT/mTOR Signaling Pathway. Cells.      2019 May 18;8(5). pii: E477. doi: 10.3390/cells8050477..

Long non-coding RNA slincRAD      functions in methylation regulation during the early stage of mouse adipogenesis.      RNA Biol. 2019 Jun 19:1-13. doi: 10.1080/15476286.2019.1631643.

Author Response

    The current study has identified novel candidate regulators that may be assessed in future molecular studies on adipogenic conversion of muscle cells.

    Skeletal muscle has recently been identified as an organ that produces and releases cytokines, which have been named “myokines”. Given that skeletal muscle is the largest organ in the human body, our discovery that contracting skeletal muscle secretes proteins sets a novel paradigm: skeletal muscle is an endocrine organ producing and releasing myokines in response to contraction which can influence metabolism in other tissues and organs [1].

    Long noncoding RNAs (lncRNAs) play a major role in adipogenesis. However, differential expression profiles of lncRNAs in inguinal white adipose tissue (iWAT) between wild-type (WT) and ob/ob mice, as well as their roles in adipogenesis, are not well understood [2]. A Novel lnc-RNA, Named lnc-ORA, Is Identified by RNA-Seq Analysis, and Its Knockdown Inhibits   Adipogenesis by Regulating the PI3K/AKT/mTOR Signaling Pathway [2].

    Up-regulation of long noncoding RNA slincRAD expression was found to occur in the early differentiation stages of 3T3-L1 cell, prior to the regulation of major transcription factors [3]. Long non-coding RNA slincRAD functions in methylation regulation during the early stage of mouse adipogenesis [3].

    Authors are kindly requested to emphasize the current concepts about these issues in the context of recent knowledge and the available literature. This article should be quoted in the References list.

R:   Many thanks for the your information and suggestions. New contents and the references have added in the revised manuscript.

Round  2

Reviewer 2 Report

My concerns were addressed.

Reviewer 3 Report

All the questions and queries were properly responded.